# Assessing Regional Weather’s Impact on Spinal Cord Injury Survivors, Caregivers, and General Public in Miami, Florida

**DOI:** 10.3390/ijerph21040382

**Published:** 2024-03-22

**Authors:** Danielle Hildegard Bass, Pardis Ghamasaee, Gregory E. Bigford, Mandeville Wakefield, Lunthita M. Duthely, Daniel Samano

**Affiliations:** 1Department of Neurology, Miller School of Medicine, University of Miami, Miami, FL 33136, USA; dhb55@med.miami.edu; 2Department of Public Health Sciences, Miller School of Medicine, University of Miami, Miami, FL 33136, USA; pxg444@med.miami.edu (P.G.); lduthely@med.miami.edu (L.M.D.); 3Miami Project to Cure Paralysis, Department of Neurological Surgery, Miller School of Medicine, University of Miami, Miami, FL 33136, USA; gbigford@med.miami.edu; 4Edward Via College of Osteopathic Medicine, Auburn, AL 36832, USA; msofer@vcom.edu; 5Department of Obstetrics, Gynecology & Reproductive Sciences, Miller School of Medicine, University of Miami, Miami, FL 33136, USA

**Keywords:** climate change, extreme weather events, knowledge-attitudes-behaviors, environmental health, disability care, risk communication

## Abstract

(1) Background: Climate change is increasing the already frequent diverse extreme weather events (EWE) across geographic locations, directly and indirectly impacting human health. However, current ongoing research fails to address the magnitude of these indirect impacts, including healthcare access. Vulnerable populations such as persons with spinal cord injury (pSCI) face added physiologic burden such as thermoregulation or mobility challenges like closure of public transportation. Our exploratory research assessed commute and transport to healthcare facilities as well as the knowledge, attitudes and behaviors (KAB) of pSCI regarding EWE and climate change when compared to pSCI caregivers (CG) and the general public (GP). (2) Methods: A KAB survey was employed to conduct a cross-sectional assessment of pSCI, CG, and GP in Miami from October through November 2019 using snowball sampling. Descriptive and logistic regression statistical analyses were used. (3) Results: Of 65 eligible survey respondents, 27 (41.5%) were pSCI, 11 (17%) CG, and 27 (41.5%) GP. Overall, pSCI reported EWE, particularly flooding and heavy rain, affecting their daily activities including healthcare appointments, more frequently than CG or GP. The overall models for logistic regression looking at commute to and attendance of healthcare appointments were statistically significant. pSCI self-report being less vulnerable than others, and a large proportion of each group was not fully convinced climate change is happening. (4) Conclusions: This study provided insight to the KAB of 3 population subgroups in Miami, Florida. pSCI are significantly more vulnerable to the effects of regional weather events yet exhibit disproportionate self-perception of their vulnerability. Continued and more comprehensive research is needed to characterize the barriers that vulnerable populations face during weather events.

## 1. Introduction

The Earth’s rapidly changing climate represents the challenge of our lifetime, and a top priority for public health professionals as there are countless implications on human health [1,2,3,4,5]. It is known that the intensity and frequency of extreme weather events (EWE) such as extreme heat, floods, and heavy rain are increasing due to climate change [6,7,8,9,10]. Research has also shown that EWE are associated with health complications such as malnutrition, heat-related illness, cardiovascular failure, and death [2,6,11,12]. However, the indirect impacts on health such as access to healthcare are rarely considered, evident in the lack of literature regarding this topic. This growing issue is particularly important for vulnerable populations, such as persons living with spinal cord injuries (pSCI) [12,13,14]. This population subgroup is likely disproportionately impacted by EWE; directly due to thermoregulation impairments and indirectly due to their transportation and physical mobility limitations which make them dependent on others [12,14,15,16,17]. 

SCI is defined as a damage to any part of the spinal cord or nerves at the end of the spinal canal, which can often cause permanent disabilities [18]. Disability-adjusted life years, or DALYs, is an international measure originally developed by the World Health Organization in 1993, that has been adopted globally in over 200 countries. This measure compares the burden of disability in years of life lost, in addition to years living with a disability [19]. Individuals living with SCI are among those with the greatest DALYs in the United States, second to those with brain cancers. This finding places the United States’ national burden of SCI above other chronic and debilitating conditions such as HIV, Parkinson’s Disease, Multiple Sclerosis, and congenital heart abnormalities [19]. Further, due to the nature and anatomic level of their injuries, pSCI are commonly dependent on caregivers (CG) and public transportation for their activities of daily living, such as commuting to and from healthcare appointments. Living in flat cities such as Miami, Florida allows pSCI to more easily self-transport, though changing climatological conditions may severely limit this ability. 

As of 2023, there were approximately 302,000 pSCI in the United States, with nearly 18,000 spinal cord injuries occurring each year [20]. Due to the potential impact climate change can have on their lives, it is important to understand this populations’ knowledge, attitude, and behaviors (KAB) regarding climate and EWE, their vulnerability, and their autonomy. The objectives of the present study are to understand the KAB of pSCI regarding climate change and EWE’s effect on their lives, such as commute to/from healthcare. pSCI were compared to SCI CG and the general public (GP). We hypothesized that pSCI are more vulnerable than their counterparts to the effects of climate change and EWE, that they do not conceptualize or understand these terms, and do not feel responsible to take actions to address their effects. 

## 2. Materials and Methods

### 2.1. Design 

This is an observational, descriptive, cross-sectional study. It was implemented pre-COVID19 pandemic, from October to November 2019. An exhaustive literature review was conducted with the goal of identifying a survey or related instruments assessing Climate and EWE’s influence on health. At the time of our survey development, only six publications were found in the literature that reported few implemented questions regarding these topics [2,21,22,23,24,25]. We developed a survey to understand population subgroups’ KAB regarding climate change and regional EWE in Miami-Dade County (MDC). Through use of a KAB survey we gathered information from populations with similar attributes and compared to other population subgroups. This type of survey is particularly beneficial to assess knowledge, attitude, and behaviors, as well as risks associated with Climate change and extreme weather events [26,27]. To avoid biasing the survey respondents, definitions of extreme weather and climate change were not provided within the survey. The survey was made available in-person and online, to ensure accessibility and ease of participation for all respondents. Given the large percentage of Hispanic/Latinx community members in MDC, the survey was distributed in both English and Spanish. Utilizing random snow-ball sampling we recruited participants at the Miami Project to Cure Paralysis, outpatient rehabilitation clinics for SCI in the largest medical center for South Florida, and in the community at local coffee shops and metro stations. Upon initial in-person contact and survey completion, participants were given a physical copy and/or link to the online version of the survey. This provided them with the study team’s contact information and inclusion criteria described below. Participants were asked to provide this physical copy or online link to others in their network. The study was approved by the Academic Institutional Review Board with a waiver for signed informed consent. 

Participants qualified for the study if they self-identified as being 18 years or older. This age criterion was uniformly applied across all participant groups (pSCI, CG, and GP). Participants also acknowledged having lived or worked in MDC for a total of six months or more prior to participating in the study. Members of the pSCI group included paraplegia and quadriplegia, recognized as S14.109A in the International Classification of Diseases, Tenth Revision (ICD-10) [28]. Demographic information, commute and transportation preferences, as well as targeted KAB questions regarding climate and health, were collected. 

For this study, the dyad concept pertains to the dynamic relationship between pSCI and CG, where the well-being and experiences of one member are intricately linked to those of the other. This is often characterized by mutual care, emotional support, or interaction. We aim to gain a deeper understanding of the challenges, coping mechanisms, and support systems within this unique relational context. Therefore, the decision was made to study pSCI and CG as two individual groups, separating them from this dyad concept. 

### 2.2. Statistical Analysis 

All demographic and continuous variables were analyzed through descriptive statistics, and frequency tables were calculated for all categorical variables. These variables were compared between the three categories: pSCI, CG, and GP. Chi-square analysis was conducted to determine significant differences by group. Missing responses for Table 1 were reported. We employed a complete case analysis approach for the remaining data to ensure conclusions were drawn from complete data. Binomial logistic regression was performed to obtain Odds Ratios for the influence of regional EWE (Extreme Heat, Flood, and Heavy Rain) on commute interruptions and outpatient healthcare attendance by pSCI, CG, and GP. *p*-values were obtained and considered statistically significant at *p* < 0.05. All analyses were performed using SAS OnDemand for Academics Version 9.4 statistical software (SAS Institute Inc., Cary, NC, USA), R studio (PBC, Boston, MA, USA) and Microsoft Excel version 16.54 (Microsoft, Redmond, WA, USA).

## 3. Results

### 3.1. Demographics

A total of 65 eligible participants completed the survey. Of these, 27 (41.5%) identified as pSCI, 11 (17%) as CG, and 27 (41.5%) as GP. Demographic information is summarized in Table 1. 

### 3.2. Commute and Transportation

Participants were asked to rank their preferred method of transportation and how frequently their commute to and from healthcare appointments was interrupted by EWE. Results are provided in Figure 1A and B, respectively. Across all groups, SCI 82%, CG 100%, and GP 85%, a majority reported a commute time to healthcare facility of less than 1 h. pSCI primarily prefer to drive themselves, however they were the only group prioritizing use of public transport and use of specialized county/city provided transport. Binomial logistic regression analyses were conducted to investigate the impact of EWE: extreme heat, flood, and heavy rain, on commute and attendance patterns among individuals with SCI compared to non-injured individuals. The analyses examined three levels of impact: none (baseline), a lot, and somewhat, for commute; and none (baseline), reschedule, and cancel/no show, for attendance (Figure 2). The impact of EWE on commute patterns revealed a significant overall model (*p* < 0.001). Within the commute model, extreme heat, whether *a lot* (OR = 0.97, 95% CI [0.22, 3.77]) or *somewhat* (OR = 1.90, 95% CI [0.45, 7.73]), did not significantly influence commute patterns for individuals with SCI compared to non-injured counterparts. However, experiencing flooding, both *a lot* (OR = 7.4, 95% CI [1.00, 69.81]) and *somewhat* (OR = 3.60, 95% CI [0.95, 14.02]) strongly impacted commute patterns for this population (significant at *p* < 0.01). Conversely, heavy rain, whether *a lot* (OR = 2.23, 95% CI [0.39, 11.61]) or *somewhat* (OR = 1.52, 95% CI [0.40, 5.51]), did not have a significant effect on commute patterns between groups. The impact of EWE on attendance also demonstrated a statistically significant overall model (*p* = 0.001). Within the attendance model, extreme heat leading to *reschedule* (OR = 0.11, 95% CI [0.01, 0.74]) strongly affected attendance patterns (significant at *p* < 0.01) but extreme heat leading to *cancel/no show* (OR = 0.09, 95% CI [0.00, 1.2]) did not significantly affect attendance patterns among individuals with SCI compared to non-injured individuals. Floods leading to *reschedule* (OR = 1.92, 95% CI [0.54, 7.91]) and *cancel/no show* (OR = 3.96, 95% CI [0.55, 27.57]) did not significantly affect attendance patterns for this population. Heavy rain leading to *reschedule* (OR = 3.46, 95% CI [1.14, 10.88]) significantly affect attendance patterns among individuals with SCI compared to non-injured individuals, but heavy rain leading to *cancel/no show* (OR = 5.24, 95% CI [0.43, 138.15]) did not have the same affect. *(Model summary statistics*, *Appendix A).*

### 3.3. Climate

To assess the self-perceived level of vulnerability, participants were asked how vulnerable they were to extreme heat, flooding and heavy rain. As well, pSCI, CG and GP were asked how vulnerable other pSCI are to the same EWEs. In the context of this study, we aimed to assess self-perceived physical vulnerability. This encompasses the physical limitations and challenges faced by pSCI when coping with EWE, including limited mobility, accessibility issues, and health-related vulnerabilities. Furthermore, the survey asked how convinced participants were that climate change is happening, and if they felt responsible to make changes to slow down climate change. Results to these questions are presented in Figure 3A, B and C, respectively.

Participants were requested to state health effects of climate change and exposure to extreme heat while outdoors. These responses were grouped into themes and represented by word charts shown in Figure 4. Overall, few participants (N = 14 (22%) across all groups) answered in response to what they think are health effects of climate. Participants who were pSCI also voiced concerns about water damage to their wheelchairs, such as: “flooding damaging my wheelchair” or “electrical outages reducing battery power”. 

Additionally, forced migration was assessed when asked whether climate change would force participants to move from their current place of residence to another place, those living with SCI (23%), CG (9%) and the general public (23%) responded yes. When asked if others would be forced to move to Miami-Dade County, SCI (19%), CG (9%), and GP (23%) stated yes.

## 4. Discussion

This study aimed to understand the KAB of pSCI regarding climate change and how EWE affect their lives. The perceived effects of EWE, including extreme heat, flooding, and heavy rain, are considerably more prevalent in pSCI than their CG or GP counterparts. pSCI reported more frequent interruptions to daily activities and more often chose to cancel healthcare appointments. This group also depended more on others for transportation to healthcare visits and described more direct health impacts when thinking of exposure to EWE. Despite being the most affected by EWE, pSCI did not perceive themselves as the most vulnerable group. Only 30% of pSCI reported being completely convinced that climate change is happening. 

MDC is recognized for spearheading resilient adaptations in response to climate change, such as the first city in the world to designate a Chief Heat Officer as a governmental office [29]. MDC’s subtropical climate, flat surfaces and diverse population is appealing to many globally, particularly to pSCI. It is comprised of 2.7 million inhabitants, of which 68% (1.87 mil) are Hispanic/Latinx individuals, therefore MDC provides familiarity for those coming from Latin America and the Caribbean [30]. Additionally, MDC is home to the Miami Project to Cure Paralysis, a world known scientifically translational institution that offers pSCI the possibility of community-based resources and support, cutting edge research, and longitudinal continuity of care. Therefore, conducting this study in MDC provided an ideal population from which to sample. Our study reflects the literature, where pSCI were predominately White (54%), Male (81%), and single (52%). In contrast, CG were primarily Female (73%) [31,32]. To best discuss the multitude of topics addressed in this study, we are presenting our discussions in the subsequent sections below. 

### 4.1. Transport Methods and Commute to/from Healthcare Appointments

It is important to consider that the level of spinal cord injury is directly related to the independence level. As an example, a neck or cervical-level motor injury is more likely to limit the physical transfer ability from a wheelchair to a car, more so to drive an adapted vehicle. Whereas a thoracic or lower back-lumbar injury is more likely to preserve upper extremity and hand function, allowing transfers and likelihood of driving an adapted car. Considering this, one of the limitations of our results are not fully understanding the differences based on level of injury. However, we do feel the generalizability of our results are important to the SCI community as a whole. Despite having a similar number of cars and people per household among pSCI, CG and GP; preferred transportation methods varied for pSCI when compared to their CG and GP counterparts. While all groups ranked driving themselves as their primary choice most frequently, pSCI were the only group to report use of public transportation, county/city provided transport, and insurance provided transport. Although pSCI were the only group to report utilization of these methods, it was only 19% of this group. 

When asked how often EWE interrupt their daily activities, pSCI also reported more frequent monthly interruptions, the most frequent being 1–5 times a month, whereas the GP reported yearly interruptions, most frequent being 1–5 a year. pSCI were also the only group to report having their daily activities interrupted by an EWE more than 10 times per month. Our logistic regression analysis aimed to predict whether EWEs would have an impact on commute and attendance to healthcare appointments among pSCI and non-injured groups. The significant impact of flooding on commute patterns suggests that individuals with SCI may face greater challenges in transportation and accessibility during flood events. This finding calls into question the need for targeted interventions and infrastructure improvements to support the mobility of individuals with SCI in adverse weather conditions. Regarding attendance patterns, extreme heat and heavy rain emerged as a significant factor contributing to rescheduled appointments among individuals with SCI. This highlights the vulnerability of individuals with SCI to extreme heat and the importance of proactive measures to mitigate the impact on healthcare access and service utilization.

Contrary to expectations, flooding did not significantly influence attendance patterns among pSCI, although heavy rain was a significant factor for pSCI, suggesting that factors beyond weather conditions, such as access to adaptive transportation and support systems, may also play an important role in determining mobility and healthcare access for these individuals during EWEs. Overall, the results emphasize the complex interplay between weather events, individual characteristics, and environmental factors in shaping commute and attendance patterns among individuals with SCI. Future research should explore additional determinants of mobility and healthcare access to inform targeted interventions and policies aimed at improving outcomes for this population.

Although our results yielded all groups’ commute times to and from their healthcare facility to be less than an hour, this does not account for the preparation time before pSCI start their commute, which in general requires planning, packing, and various safety precautions [16,18,33]. In addition, it does not account for the wait time to be picked up both at home to attend healthcare and vice versa. This may demonstrate an underreported commute time and/or interruptions from those living with SCI. Thus, the vulnerability of pSCI is clear as their daily transportation habits are dependent on others, despite Miami, FL being a relatively flat city that could increase independent transport opportunities [34,35,36,37]. 

### 4.2. Climate

#### 4.2.1. Vulnerability

Only 19% of pSCI self-reported as the most vulnerable to the effects of climate change when asked about other groups such as the elderly or people living with HIV. Interestingly, up to 37% of pSCI respondents reported other individuals living with SCI as more vulnerable than themselves. There is a disconnection between pSCI’s perception of self-described vulnerability vs how they perceived other pSCI. The GP reported pSCI being almost four times more vulnerable to the effects of EWE, whereas CG thought pSCI are less vulnerable than any other group. What is unique about the SCI population is that everyone’s ability is different, independent of their level of injury. Within the vulnerability faced by the physiology of SCI, having a higher level of injury (for example, cervical versus lumbar level), makes the individual more dependent on others. These individuals would therefore likely be more vulnerable than their paraplegic counterparts. In this study, we did not separate by level of injury, therefore, the responses about others living with SCI may not necessarily relay a discrepant self-vulnerability belief but may also refer to other SCI levels of injury. We did not collect data regarding level of injury, as two individuals with the same injury level could still vary significantly in terms of vulnerability. Similarly, their reported vulnerability could be influenced by external factors such as access to healthcare, insurance status, support network, and time since injury.

#### 4.2.2. Conviction in Climate Change

The Earth’s climate is changing, including an increase in the global surface temperature by over 1 °C faster in the last 50 years than in any other period [38]. South Florida and Miami in particular, has been named “Ground Zero” for climate change [39]. However, of those in our sample who answered the question “how convinced are you that Climate Change is happening?”, (30%) were pSCI, (12%) were CG, and (61%) were GP, who were completely convinced it is happening. Therefore, 70% of pSCI, 88% CG, and 39% GP are mostly convinced, not convinced or don’t know what climate change is. This could be explained by comparing responses to self-reported highest achieved education level. Those living with SCI reported having completed high school or less 43% of the time and CG 46%. This contrasts with GP, who reported having completed some college or more 89%, with 48% of them completing graduate school. Given the large number of the GP reporting higher education achievements, it may be implied that a higher education status leads to better understanding and knowledge regarding global climate change [40]. 

#### 4.2.3. Responsibility to Make Changes to Slow Down Climate Change

Despite varying degrees in conviction of climate change, 54% of pSCI, 87% of CG, and 77% of GP reported feeling responsible to make changes to address its effects. Efforts to make changes that could mitigate the effects of climate change could take many forms. Our study sample reported a majority feel responsible to do so, yet we hypothesize they likely don’t know how to make these changes. Local government initiatives should utilize community programming that could educate individuals as to how they can best help. 

At a population level, these findings point towards an area of opportunity where Public Health efforts can communicate reliable information to all public population subgroups, not only the most vulnerable. The authors believe that all population subgroups are vulnerable to climate change and extreme weather events, the difference lies in the level of vulnerability by disabilities and social determinants of health. These social determinants are shaped by various social, economic, and environmental factors that impact access to resources, opportunities, and support systems [41]. 

#### 4.2.4. Forced Migration

Our findings suggest that all groups have somewhat levels of belief that climate change would force them to move to a different location pSCI (23%), CG (9%) and GP (23%), and also believe others will move to MDC due to same reasons pSCI (19%), CG (9%), and GP (23%), respectively. Projections indicate that MDC will be significantly impacted by flooding and increasing temperatures in the subsequent 3 decades [42]. These discrepancies are reflective of the uncertainty that we all live nowadays, where we know climate change is happening and have proof of it, but predictions are still too uncertain, more so at a regional weather level such as MDC. 

### 4.3. Dyad

Our approach to the dyad concept serves as a foundational framework for exploring the complex interactions, shared experiences, and mutual influences between pSCI and CG. Understanding their dynamic is crucial to comprehensively assess the impact of spinal cord injury on both individuals and for developing targeted interventions to support their needs effectively.

While one could assume that pSCI and CG would have similar perceptions and responses due to their close relationship to one another [16,43,44,45], our findings showed variations between these two subcategories, making it vital to separate them to gather a better characterization of each group. Those living with SCI have great concern for flooding and report it to be an interruption resulting in their not attending healthcare appointments. pSCI reporting specific concerns of “flooding [destroying] my wheelchair”, lack of electricity limiting access to vital health needs, and “ruining my electric wheelchair from water damage.” However, their CG were found to perceive flooding to not be a large issue when it comes to attending medical appointments. This variation could be due to the difficulties faced by those living with SCI in terms of thermoregulation during heavy rain if getting wet and transportation or commute difficulties in case of flooding [15,26,46,47]. Those who are CG of those living with SCI do not necessarily have to face these difficulties themselves, and therefore, are prone to not take into consideration the dangers of flooding for their own day to day healthcare appointments. This incongruency of thought between pSCI and CG come despite them frequently cohabitating, with pSCI more often relying on their CG for day-to-day activities. Future interventions need to address this incongruency through educating and informing both pSCI and CG, to make sure CG better address and care for pSCI and their needs.

Looking further at the lifelong burdens faced by those with SCI, it is important to consider their disability-adjusted life years, or DALYs. A study focused on the United States looked at 53,544 cases of SCI and calculated the DALYs associated with their injury. Overall, those with cervical SCI accounted for the majority of DALYs amongst those with SCI. In total, during their study period, those living with SCI accounted for 1,066,499 DALYs or years of health life lost, making SCI the second highest cause of DALYs in the US [19]. Our findings add to potential causes of DALY accumulation for those with SCI, as the difficulties faced with EWE leads to cancelation and lack of adherence to their healthcare plans physical therapy and other, causing detrimental health effects, thus potentially loss of healthy life years. We expect the number of DALYs to continue to grow, as the number of EWE increases due to climate change, consequently leading to more indirect effects on healthcare access such as canceled appointments. 

### 4.4. Limitations

Limitations for this study include those of a cross-sectional study design. These include confounding variables (i.e., demographics and socioeconomics) and the lack of longitudinal data, which makes it impossible to establish a true cause and effect relationship [48]. Given the exploratory nature of our research and our primary focus on public health questions, we intentionally chose not to delve into statistical interactions or interpretations that may detract from the clarity and simplicity of our findings. As well, participants were asked to recall information regarding their past year of climate-related transportation habits, therefore recall bias must be considered [48]. Further, due to the lack of previously powered studies, we recognize the sample size for this preliminary study is smaller than ideal for logistic regression, in particular. As such, we are unable to make vast generalizations regarding those living with SCI, CG, and the GP in Miami, Florida [49]. Further, it was up to the participants discretion whether to respond to certain questions, therefore, there are a varying number of responses to each question. This variation could impact the significance of variables seen in the chi-square and binomial logistic regression analyses. Larger-scale studies are needed to better characterize these associations. Future studies should characterize by level of injury- cervical, thoracic, lumbar- and stratify behaviors such as driving, commute, as higher levels of injury would make driving more limited and more dependent on others.

## 5. Conclusions

Our study reports exploratory data to support further comprehensive investigations into the broader determinants of EWEs on commute interruptions and attendance among individuals with SCI and their caregivers. Reproducible prediction models are the future of climate and health research as varying climates are present at different geographic regions. However, to better characterize future dynamic weather patterns, we must first understand both the past observations by population subgroups, in this case pSCI, and assess their KAB to identify potential areas for public health interventions, from the individual to the policy levels. This study provided evidence that while those living with SCI are known to be more vulnerable to the effects of climate change, their self-perception of their vulnerability is skewed. Additionally, regardless of population subgroup, no one was fully convinced about climate change. Our results also indicate that EWEs significantly impact commute to and attendance to healthcare appointments among pSCI. These findings point to areas of opportunity to develop public health interventions to educate and influence the behaviors of pSCI, CG and GP. In order to influence policy that addresses population subgroups, specifically pSCI, more research is needed in the field of EWE, climate and health. We recognize the potential for future research to explore these factors in greater depth, incorporating multivariable analyses and controlling for demographic and socioeconomic variables to elucidate their impact on the outcomes observed in our study.

## Figures and Tables

**Figure 1 ijerph-21-00382-f001:**
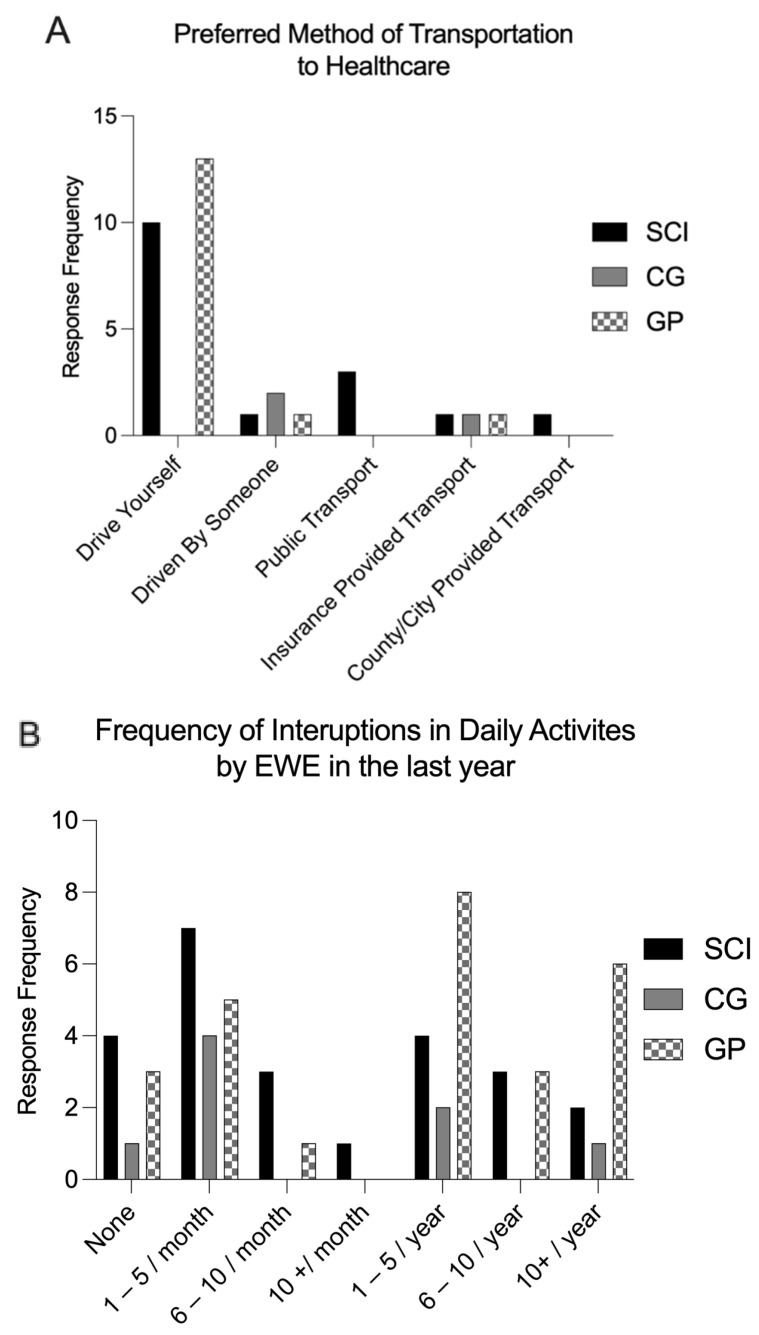
(**A**) Preferred method of transportation to healthcare by pSCI, CG and GP. (**B**) Daily activities interruptions by Extreme Weather Events in pSCI, CG and GP in the last year. This refers to the 12-month period immediately preceding the date the survey was completed by each participant. Response frequency indicates the number of responses received for each option.

**Figure 2 ijerph-21-00382-f002:**
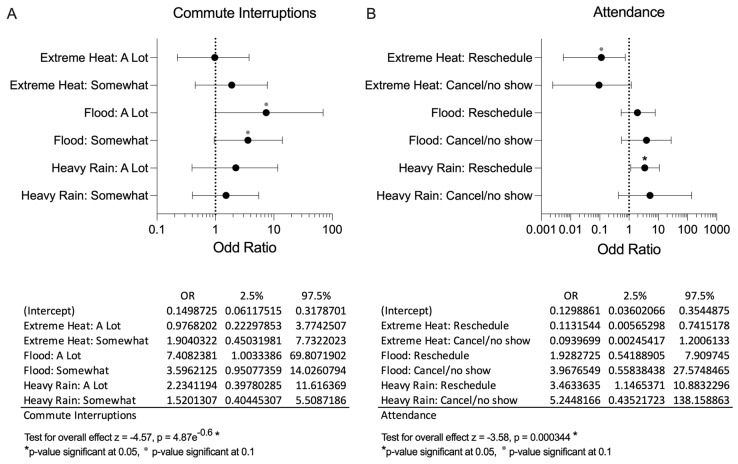
Binomial logistic regression. Likelihood of Extreme Weather Events influencing healthcare utilization by pSCI when compared with CG and/or GP. This model reflects all groups in the study sample. It demonstrates how pSCI group is more likely to experience commute interruptions (**A**) and not attend Healthcare appointments (**B**) following each weather event.

**Figure 3 ijerph-21-00382-f003:**
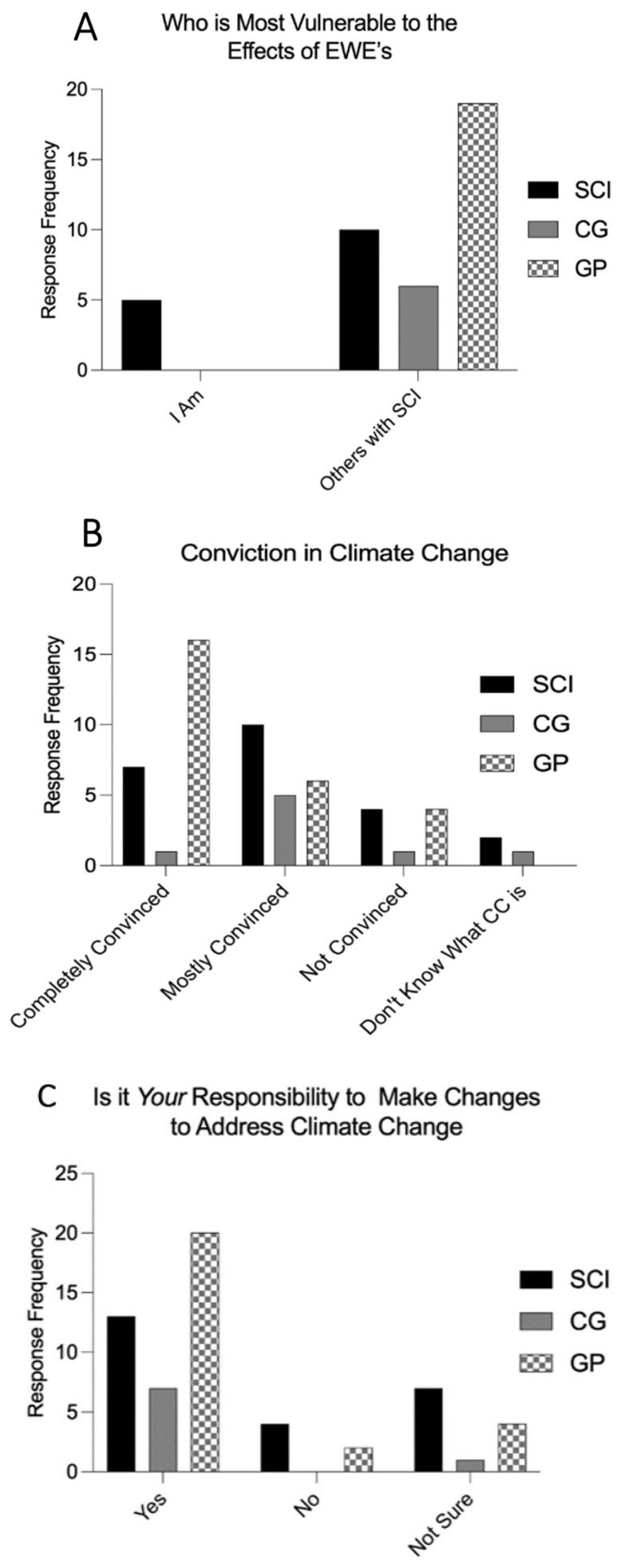
(**A**) Perceived level of vulnerability to EWE for pSCI. Those with SCI reported for themselves and other SCI population groups. (**B**) How convinced are you that Climate Change is happening? By pSCI, CG and GP. (**C**) Are you responsible to make changes to slow down climate change? By pSCI, CG and GP.

**Figure 4 ijerph-21-00382-f004:**
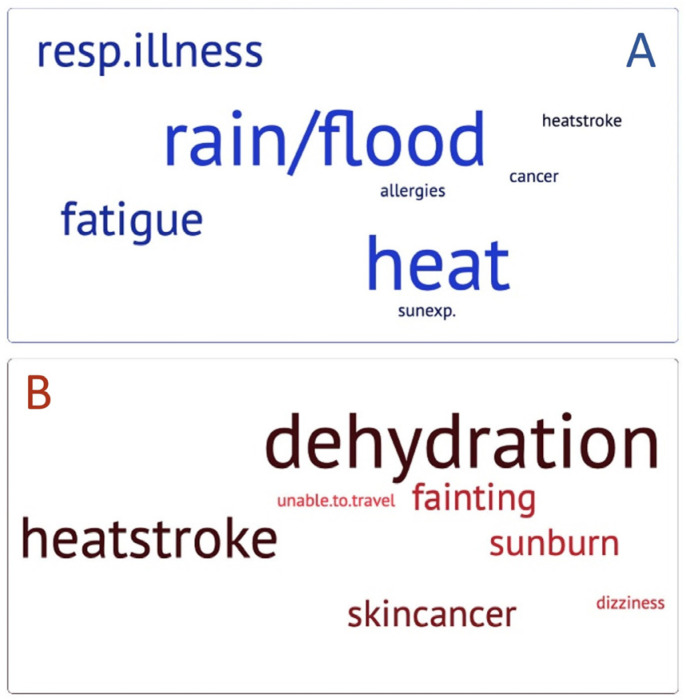
Stated health effects of climate change in blue (**A**) and exposure to extreme heat while outdoors in red (**B**). The larger the word, the more frequently it was seen and reported in the responses.

**Table 1 ijerph-21-00382-t001:** Demographics. pSCI–People Living with Spinal Cord Injury, CG–Caregiver, GP–General Public.

	pSCI	CG	GP
	Mean	Mean	Mean
Age (years)	41.9	46.9	35
	Frequency (%)	Frequency (%)	Frequency (%)
Gender			
Male	22 (81%)	3 (27%)	17 (63%)
Female	5 (19%)	8 (73%)	10 (37%)
Race			
White	13 (48%)	7 (64%)	22 (81%)
African American/Black	6 (22%)	3 (27%)	1 (4%)
Native American	1 (4%)	0 (0%)	0 (0%)
Do Not Wish to Answer	4 (15%)	0 (0%)	4 (15%)
Missing	3 (11%)	1 (9%)	0 (0%)
Ethnicity			
Hispanic	17 (63%)	6 (55%)	20 (74%)
Non-Hispanic	7 (25%)	2 (18%)	2 (7%)
Haitian	1 (4%)	1 (9%)	1 (4%)
Other Caribbean Islander	1 (4%)	1 (9%)	0 (0%)
Do Not Wish to Answer	1 (4%)	1 (9%)	3 (11%)
Missing	0 (0%)	0 (0%)	1 (4%)
Marital Status			
Married	8 (29%)	3 (27%)	12 (44%)
Single	14 (52%)	5 (46%)	10 (37%)
Divorced	4 (15%)	2 (18%)	1 (4%)
Living with Significant Other	1 (4%)	1 (9%)	4 (15%)
Missing	0 (0%)	0 (0%)	0 (0%)
Highest Level of Education			
Less than 7 years of school	0 (0%)	1 (9%)	0 (0%)
Junior High	1 (4%)	0 (0%)	0 (0%)
Some High School	3 (11%)	0 (0%)	0 (0%)
High School Graduate	7 (26%)	4 (37%)	3 (11%)
Some College	5 (18%)	3 (27%)	2 (8%)
Completed College	11 (41%)	2 (18%)	9 (33%)
Completed Graduate School	0 (0%)	1 (9%)	13 (48%)
Missing	0 (0%)	0 (0%)	0 (0%)
Annual Household Income			
Less than 20,000	10 (37%)	5 (46%)	2 (8%)
21,000 to 55,000	6 (22%)	2 (18%)	3 (11%)
56,000 to 96,000	6 (22%)	3 (27%)	3 (11%)
Greater than 96,000	3 (11%)	1 (9%)	19 (70%)
Missing	2 (7%)	0 (0%)	0 (0%)
How long is the commute/travel to your health care facility?			
0–30 min	13 (48%)	5 (45%)	18 (67%)
31–60 min	9 (33%)	6 (55%)	5 (18%)
1–2 h	5 (19%)	0 (0%)	4 (15%)
Missing	0 (0%)	0 (0%)	0 (0%)
	Mean	Mean	Mean
Number of Cars in household	1.4	0.9	1.6
Number of People in your household	2.8	2.6	2.6

## Data Availability

The data presented in this study are available on request from the corresponding author.

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
