# Peer review of "Assessing Regional Weather’s Impact on Spinal Cord Injury Survivors, Caregivers, and General Public in Miami, Florida"

_ijerph, 2024, doi:10.3390/ijerph21040382_

Round 1

Reviewer 1 Report

Comments and Suggestions for Authors

I commend the authors for their rigorous methodology and the coherent organization of the manuscript. However, I do have a few suggestions for potential improvements:

Manuscript title

-       Ensure that the manuscript title emphasizes important information, as it is currently too long.

Abstract

-       Please provide details regarding the study period, study area, and the sampling methods employed.

-       It is important to include the findings obtained through logistic regression analysis.

-       Ensure that the selected keywords do not overlap with the manuscript title.

Introduction

-       Regarding line 64-65, As of 2020, there were approximately 300,000 pSCI in the United States..., please ensure the information is updated if necessary.

-       In line 70, where caregivers (CG) is mentioned, please review the entire manuscript to ensure that the full name is not repeated when the acronym has already been introduced in the previous paragraph.

Materials and Methods

Design

-       Include a map of the study area, illustrating the sample size and representing the population under study.

-       In lines 95-97, where it mentions, Participants qualified for the study if they self-identified as being 18 years or older..., please clarify if this criterion applies to both the group of caregivers and the general public.

-       In lines 97-98, where it states, The study sample consisted of: pSCI, those who identify as SCI-caregivers (CG), and the general public (GP)., please provide the ICD-10 code for pSCI. Additionally, ensure that the full names are not repeated unnecessarily throughout the manuscript.

-       In line 101, where it mentions the dyad concept, please provide clarification to ensure readers understand its significance within the context of the study.

Statistical Analysis

-       Please specify the dependent and independent variables used in the study.

-       In lines 108-109, where it mentions High Heat Index and Heavy Rain, please provide an explanation or definition to clarify these terms within the context of the study.

Results

Demographics

-       Regarding lines 114-115, where it states, A total of 65 eligible participants completed the survey. Of these, 27 (41.5%) identified..., please reconcile any discrepancies between these numbers and those reported in the Abstract.

-       Table 1, please verify the participant counts for each variable across the three groups of participants.

Commute and Transportation

-       In Figure 2A, please clarify what the response frequency represents, specifically the frequency per what unit or category.

-       In Figure 2B, please specify the year referred to when mentioning EWE in the last year.

Climate

-       In line 143, where it mentions the level of vulnerability, please provide further explanation or elaboration to clarify the concept within the context of the study.

Discussion

Responsibility to make changes to slow down climate change

-       In line 270, where it mentions social determinants of health, please provide an explanation to elucidate this concept further.

Forced Migration

-       In lines 275-276, where it states Projections indicate that MDC will be significantly impacted by flooding and increasing temperatures in the subsequent 3 decades.”, please ensure to include the appropriate reference to support this statement.

Limitations

-       In line 315, (wang), line 317, (Wang et al.), and line 319, (Jones et al.), please verify the correctness of the citation format.

Reviewer 2 Report

Comments and Suggestions for Authors

The study addresses a current and relevant issue. Overall, the manuscript was presented in a clear and intelligible fashion. However, the methods require improvement.

The survey instrument was researcher-structured but there were no details about the resources used and content of the instrument.

The sample size for this study was rather small and reduced the power of the study. How was this sample size determined? The pSCI are a heterogenous group as has been highlighted in the discussion; the selected group does not appear representative. Specifically, no information was provided about the level of cord injury, a factor that significantly determines the degree of disability. 

Were there missing data? How was missing data handled?

The logistic regression analysis is inappropriate and should be removed; a sample size of barely above 50 would result in imprecise effect sizes as was the case in this study. 

The drawbacks of a cross-sectional design was correctly highlighted. Important variable and potential confounding factors were not considered and so inferences made are rather weak. 

Results

In Table 1 , No. of cars  and No. of people should be presented as frequency and percentages not mean (SD); each variable can be categorised and percentages provided.

The figures suggest that there were missing data or non-responses; the frequencies do not add up. For instance, Figure 2a- The percentage of CG who preferred to drive themselves was not shown. Fig 2b-It is unclear if this was a multiple response question, the frequencies do not add up.

Discussion

This was fairly well written and backed by the findings, however the lack of some important information weakens the argument. There was minimal background information and critical appraisal of similar existing studies.

Comments on the Quality of English Language

The English Language quality was good. However, there are few unclear sentences, particularly in the results section. 

Reviewer 3 Report

Comments and Suggestions for Authors
  1. 1. The overall organization of this paper requires improvement. The discussion section, in my view, is not the suitable location to introduce related concepts such as DALY. Instead, these concepts should be integrated into the research question or outcomes.

  2. 2. Secondly, despite the inclusion of descriptions regarding demographic and socioeconomic characteristics, the author does not appear to incorporate these factors into their model. The presentation mainly consists of odds ratios supporting the author's argument. However, a more comprehensive understanding would be gained by controlling how other demographic and socioeconomic factors influence commute interruptions or attendance.
